# Characteristics of the (Auto)Reactive T Cells in Rheumatoid Arthritis According to the Immune Epitope Database

**DOI:** 10.3390/ijms24054296

**Published:** 2023-02-21

**Authors:** Caroline Carlé, Yannick Degboe, Adeline Ruyssen-Witrand, Marina I. Arleevskaya, Cyril Clavel, Yves Renaudineau

**Affiliations:** 1Referral Medical Biology Laboratory, Immunology Department, Institut Fédératif de Biologie, Toulouse University Hospital Center, 31300 Toulouse, France; 2Laboratory of Cell Biology and Cytology, Institut Fédératif de Biologie, Toulouse University Hospital Center, 31300 Toulouse, France; 3Toulouse Institute for Infectious and Inflammatory Diseases (INFINITy), INSERM U1291, CNRS U5051, University Toulouse III, 31062 Toulouse, France; 4Rheumatology Department, Toulouse University Hospital Center, 31300 Toulouse, France; 5Central Research Laboratory, Kazan State Medical Academy, 420012 Kazan, Russia; 6Institute of Fundamental Medicine and Biology, Kazan (Volga Region) Federal University, 420008 Kazan, Russia

**Keywords:** rheumatoid arthritis, memory T cells, shared epitope, neoepitopes, peptides

## Abstract

T cells are known to be involved in the pathogenesis of rheumatoid arthritis (RA). Accordingly, and to better understand T cells’ contribution to RA, a comprehensive review based on an analysis of the Immune Epitope Database (IEDB) was conducted. An immune CD8+ T cell senescence response is reported in RA and inflammatory diseases, which is driven by active viral antigens from latent viruses and cryptic self-apoptotic peptides. RA-associated pro-inflammatory CD4+ T cells are selected by MHC class II and immunodominant peptides, which are derived from molecular chaperones, host extra-cellular and cellular peptides that could be post-translationally modified (PTM), and bacterial cross-reactive peptides. A large panel of techniques have been used to characterize (auto)reactive T cells and RA-associated peptides with regards to their interaction with the MHC and TCR, capacity to enter the docking site of the shared epitope (DRB1-SE), capacity to induce T cell proliferation, capacity to select T cell subsets (Th1/Th17, Treg), and clinical contribution. Among docking DRB1-SE peptides, those with PTM expand autoreactive and high-affinity CD4+ memory T cells in RA patients with an active disease. Considering original therapeutic options in RA, mutated, or altered peptide ligands (APL) have been developed and are tested in clinical trials.

## 1. Introduction

Rheumatoid arthritis (RA) is a chronic, inflammatory, systemic autoimmune disease of the joints leading to cartilage destruction and bone erosion [1]. RA affects 0.5–1% of the population, which makes it the most common inflammatory arthropathy in Western countries. Multiple triggering factors are involved, half of which are genetic factors and half are environmental and sex-related factors, as supported by epidemiological studies and the use of Mendelian randomization approaches testing direct associations [2,3,4,5,6]. 

The RA immune response development is suspected to start several years before RA onset in the inflamed mucosa of the lung, mouth, and/or gut with the emergence of neo-epitopes from cell death and post-translational modifications (PTM), including citrullination, carbamylation, and/or acetylation [7]. This leads to neo-epitope intake by antigen presenting cells (APC), their migration into lymphoid organs, and lymphocytes’ activation with the production of anti-modified protein antibodies (AMPA), including anti-citrullinated protein antibodies (ACPA). As a consequence, an immune response can take place in the mucosa against bacterial species (e.g., *Porphyromonas, Aggregatibacter, Prevotella*) and enzymes (e.g., peptidylarginine deiminase or PAD) implicated in the neo-antigen process. These neo-antigens can be derived from mucosal bacteria membranes, histones from the uncontrolled formation of neutrophil extracellular traps (NETs), and extracellular matrix or cellular proteins [8,9,10,11]. An immune response also takes place in inflamed and arthritic joints at the preclinical stage of the disease due to the accumulation of neo-antigens present on histones and extracellular matrix proteins such as fibrinogen, collagen, tenascin, and another cell component like enolase and vimentin. The autoantibody (autoAb) spectrum in RA is completed by the presence of rheumatoid factor (RF), an anti-immunoglobulin G (IgG) autoantibody, which is associated with disease severity [12,13].

RA is considered as a prototypic CD4+ T cell disease that drives inflammation and autoAb production. CD4+ T cells are predominant in the synovial tissue of RA patients [14], and similarly to synovial CD4+ T cells, peripheral blood CD4+ T cells are able to recognize auto-antigens and to induce a cellular response. Accordingly, it has been proposed that naïve T-cell differentiation into Th1 cells is associated with the production of pro-inflammatory cytokines such as interferon gamma (IFN-γ, specific to T cells and NK cells), TNF-α, and lymphotoxin, leading to chronic inflammation and destruction of bone and cartilage [15]. However, this paradigm suffers from major limitations that include the non-response of monoclonal antibodies (mAb) against IFN-γ and CD4+ T cells in RA patients, the presence of Th17-IL-17 instead of Th1-IFN-γ positive CD4+ cells within the rheumatoid synovium, and a paradoxical IFN-γ anti-arthritis effect reported in RA-prone murine models as compared to the promotion of arthritis flares by IFN-γ in normal mice [16,17,18,19]. This Th1 model is further complicated by the report of an immunosenescence of the cytotoxic CD8+/CD4+ T cells and regulatory T cells (Treg), the pro-inflammatory contribution of Th17 cells, the presence of extra-follicular lymphoid structures in synovium, and the presence of recently described ‘peripheral helper’ T cells inducing plasma cell differentiation (in vitro) and antibodies production [20]. Considering murine models of RA, it is shown that CD4+ T cell depletion strategies abrogate T cell development, while a depletion in CD8+ T cells increases the severity of the disease [21]. In a model of severe combined immunodeficiency (SCID), mice develop arthritis after an adoptive transfer of primed autoreactive CD4+ Th1-cells followed by a challenge with the auto-reactive target [22]. Overall, these results highlight the pathophysiological importance of T cells in RA that is further reinforced by the observation that autoantigen presentation to CD4+ T cells is restricted in human to the human leukocyte antigen (HLA) class II DRB1 shared epitope (DRB1-SE) alleles and in mice to the major histocompatibility (MHC) class II H2 alleles.

Accordingly, and to improve our understanding regarding T cell functions in RA, we took advantage of the Immune Epitope Database (IEDB) to better understand the T cell responses in RA.

## 2. Immune Epitope Database

### 2.1. Data Extraction from IEDB

Given the importance of the T cell responses in RA, an RA-related antigen list was generated from the IEDB web site (http://www.iedb.org/, accessed on 7 February 2023) using the following filters: any epitope, no MHC restriction, any host, and “rheumatoid arthritis” [23]. From this analysis, and as presented in Table 1 for T cell antigens, 1390 epitopes were found corresponding to 79 antigens from various origins (human, other mammalian origin, and infectious agent), 865 assays, and 103 references. The same analysis was performed with MHC and B cell antigens. 

From this list and since not all epitopes and antigens are relevant to RA, publications were individually reviewed in depth in order to establish a list of major epitopes/antigens based on their relevance to RA. Moreover, and when available, information regarding HLA class I/II restriction, T cell activation, the phenotype of the T cell population expanded, origin of peripheral blood or synovial fluid T cell, the capacity to enter the docking site of the DRB1-SE, cross-reactivity, the importance of PTM in generating neo-epitopes, antagonist effect, and TCR molecule usage were further collected from the database and/or from publications. From these sources, two tables were generated based on HLA I/II restriction usage and antigen characteristics for the latter. To help the reader, the main techniques used to characterize RA-associated (auto)reactive T cells are reviewed below.

### 2.2. T Cell Activation

T cell activation requires two signals: first, a T cell receptor (TCR) recognition of specific peptides presented by MHC class II for CD4+ T cells or by MHC class I for CD8+ T cells, and second, simultaneous co-signals induced by CD28 or CD40L receptor engagement. Both signals can be provided by circulating monocytes present in peripheral blood or, better, after monocyte differentiation into dendritic cells (DCs) [24]. Due to the capacity of these APC to process and present relevant immunodominant peptides to T cells, full-length proteins can be used instead of peptides. Immunodominant peptide identification requires the use of overlapping and single peptides coupled with the use of in silico predictive tools.

RA-associated T cell characteristics (Table 2) can be appreciated by measuring lymphocyte proliferation after 3H-thymidine incorporation into new strands of chromosomal DNA during mitosis, flow cytometric analysis of activation markers (e.g., CD69, CD40L/CD154), and the cytokine-secreting profile using, among other techniques, enzyme-linked immunospot (ELISpot) assays. The most used cytokine marker for T cell activation is IFN-γ, but other cytokines can also be explored to investigate Th1 cell types (IFN-γ, TNF-α, IL-2, IL-6, IL-12, and IL-21), Th2 cell types (IL-4, IL-5, IL-10 and IL-13), Th17 cell types (IL-17), T regulatory (Treg) cell type (IL-10), and monocyte/macrophage activation markers such as the monocyte chemoattractant protein 1/2 (MCP-1/CCL2) and the macrophage inflammatory protein 1 alpha (MIP-1α/CCL3) [25,26].

### 2.3. T Cell Antigen Specificity

MHC multimers (usually tetramers) are valuable tools for monitoring (auto)antigen-specific T cell frequencies and phenotypes during RA development or disease progression or in response to treatments. Initially developed to follow clonally expanded antigen-specific CD8+ T cells in blood and synovial fluid [27], the technique is based on the use of streptavidin–fluorochrome conjugates associated with mono-biotinylated tetramer class I MHC-peptide complexes. For antigen-specific CD4+ T cells, the use of tetramer-class-II–peptide complexes are more challenging, mainly due to their low frequency. For this reason, dextramers, based on a dextran backbone, bearing multiple fluorescein and streptavidin moieties can be used to allow multibinding and selection of low-affinity antigen-specific T cells. The multimer technology further allows antigen-specific T cell purification.

### 2.4. T Cell Receptor Sequencing

Autoreactive clonal T cell expansion can be assessed by testing TCR repertoire diversity in peripheral blood and synovial fluid of RA patients using purified CD4+ or CD8+ T cells and TCR DNA amplification through multiplex RT-PCR or next-generation sequencing (NGS). When coupled with a single T cell analysis, TCR sequencing can be paired with a transcriptomic analysis, providing information regarding αβTCR chain association, immune phenotype, cell cycle, and metabolism [28,29]. The main limitations are related to the fact that autoreactive T cell prevalence is low and that cytokines used in long-term tissue culture can cause bias in the TCR repertoire. These effects can be limited by isolating high-affinity T cells directly ex vivo with MHC multimers but again with a risk of overlooking autoreactive T cells with lower affinity.

## 3. CD8+ T Cell Expansion

According to the established CD8+ T cell IEDB RA-list (Table 3), the main RA-associated class I epitopes are limited and derive from viral antigens and cryptic epitopes.

### 3.1. CD8 Resident Memory T Cell Inflation

In humans, an important part of the immune T cell response is directed against bacteria and viruses that produce a flurry of foreign antigens. Immunization takes place in lymphoid organs where antigen-driven naïve T cells expand and differentiate into effector and memory T cells. Next, differentiated T cells migrate through a chemokine gradient into inflamed tissue. Part of these cells can persist in the tissue as resident memory T (T_RM_) cells to provide immune protection but also to drive local inflammation [29,37,38,39,40].

Analysis of the synovial fluid and tissues from RA patients reveals an increased number of cytotoxic effector T cells and T_RM_ cells. Among them, an anti-viral response is reported in RA patients with an oligoclonal enrichment of Epstein–Barr virus (EBV)- and cytomegalovirus (CMV)-specific CD8 T_RM_ cells together with an HLA class I restriction in up to 15% in synovial fluid as compared to ~1% in peripheral blood [27,30,31,32,33,34]. In line with a persistent antiviral response fueled by viral reactivation, the synovial CD8+ T_RM_ cell response is directed against HHV active lytic proteins (e.g., BZLF1, BMLF1, pp65) rather than latent proteins (e.g., EBNA, LMP2) and non-latent viruses such as influenza [32,33,34].

The contribution of members of the human herpes viridae (HHV) family as risk factors that may trigger RA is debated, as well as any HHV contribution to disease severity [3,41,42]. However, several reports support an exacerbated HHV reactivation from their latent forms at the early stages of RA for herpes simplex virus (HSV)1/2 [43,44], and following introduction of immunosuppressive drugs for herpes zoster, CMV and EBV [45]. This reactivation does not necessarily take place in the joints, but synovial inflammation promotes the attraction and survival of the synovial pool of resident anti-HHV CD8+ T_RM_ cells [5,46]. Expansion of this pool in the joints is associated with disease activity and contributes to extra–articular involvement as CD8+ T_RM_ cells possess the capacity to migrate into inflamed tissues [18,47]. Such an effect has been named T_RM_ memory inflation and is described in numerous types of chronic inflammatory arthritis including RA [48].

### 3.2. CD8+ T Cells Immunosenescence

Latent HHV including CMV infections enhance CD8+ effector and T_RM_ cells’ immunosenescence characterized by increased capacity to release IFN-γ, loss of the CD28 co-stimulatory signals, up-regulation of the inhibitory NK cell receptor leukocyte immunoglobulin-like receptor 1 (LIR-1 or CD85), migration toward the fractalkine gradient into inflamed tissues such as the synovium due to the expression of the chemokine receptor CXCR3, and an oligoclonal TCR repertoire [30,31,49]. As a consequence, and in addition to improperly controlling latent HHV infections, CD8+ CD28-negative T cells are suspected to perpetuate the inflammatory signal with an increased burden [50].

### 3.3. CD8+ Cryptic-Apoptosis Autoreactive T Cells

During chronic viral infections, a bystander CD8+ T cell response may be directed to cryptic self-antigens generated by caspases and unveiled during cellular apoptosis [51]. Using a multimer approach to characterize them, it was reported that HLA-A2 restricted apoptotic epitopes (AE) target specific CD8+ (CD107a+) T_M_ cells [35,36]. This subset is significantly increased in RA patients, and its level is correlated with disease activity and is sensitive to anti-TNF-α therapies within RA responders. Moreover, AE-CD8+ T_M_ cells express granzyme B, produce high levels of TNF-α and selectively contact Treg and control their expansion (Figure 1).

## 4. Immunodominant DRB1-SE Antigens

### 4.1. DRB1-SE

The antigen-binding groove of the HLA class II molecule binds a limited repertoire of peptide fragments 9–15 amino acids in length, which constitutes a specific ligand for the TCR of CD4+ T cells to elicit a response. The stabilization of the peptide–HLA complex depends on four pockets (P1, P4, P6, and P9) that determine which peptides the receptor can interact with. In the third hyper-variable region of the DRB1 chain, the P4 pocket contains the SE motif at positions 70–74, and at its base the amino acids 11 and 13 that allow the binding of both negatively charged (Asp and Glu) immunodominant peptides. Pockets P1/P9 are further implicated in peptide stabilization [52].

The DRB1-SE motif accounts for the highest RA genetic risk factor with an odds ratio (OR) ranging from 2.17 (95% CI:1.94–2.42) for DRB1*01:01/02 to 4.44 (CI95: 4.02–4.91) for DRB1-SE*04.01, which benefits from the additive effect of a valine in position 11 [53]. As a consequence, and according to the population studied, 50–70% of RA patients and up to 95% of erosive RA patients possess one or two copies of the DRB1-SE, as compared to 20–50% within the healthy control group. Of note, this genotype confers higher risk for the development ACPA-positive RA disease [54].

### 4.2. DRB1-SE Restricted Immunodominant Peptides

A long list of peptides with the capacity to enter the docking site of RA-associated DRB1-SE alleles have demonstrated their ability to promote CD4+ T cell activation. According to the immunodominant CD4+ T cell RA-IEDB list (Table 4), the main DRB1-SE restricted peptides eluted from synovial tissues derive from highly conserved antigens such as molecular chaperones, synovial peptides, cross-reactive bacterial antigens, and extracellular matrix or cellular peptides with PTM modifications, mainly citrullination.

#### 4.2.1. DRB1-SE and Heat Shock Proteins

One of the best-known antigens triggering a T cell response in patients with RA is related to heat-shock proteins (HSPs). HSPs are molecular chaperones overexpressed in response to pro-inflammatory cytokines, which is the case in the synovial tissue of RA patients [64,94]. HSPs are ubiquitous and conserved proteins present in both prokaryotic and eukaryotic organisms that possess an agonist or antagonist DRB1-SE-dependent effect in their peptide structure.

The agonist or effector effect leading to T cell proliferation is carried by cross-reactive epitopes with similarities reported between the mycobacterial (myc)HSP70/DNAk peptide 287–306 (DRTRKPFQSVIADTGISVSE) and the human chaperone binding immunoglobulin protein 336–355 (BiP, RSTMKPVQKVLEDSDLKKSD) [58]. DRB1-SE effector peptides are also retrieved from GroEL2, the bacterial homolog of the HSP60; DnaJ, the bacterial homolog of the HSP4055; the 19 kDa lipoprotein (LpqH), extracted from M. tuberculosis lysate; and influenzae antigens such as hemagglutinin (Hae, 307–319) and matrix protein 1 (MP1, 17–29). The rat matrix metalloproteinase (MMP)-3-derived peptide 446–460, which presents molecular mimicry with mycHSP65 178–186, is effective in inducing T cell proliferation in PBMC from RA patients and proinflammatory cytokine release [63].

The DRB1-SE’s restricted and effector effect is counterbalanced by an antagonist or tolerogenic effect, as the BiP peptide (456–475, DNQPTVTIKVYEGERPLTKD) contributes to the proliferation of IL-10 secreting Treg cells both in vitro and in vivo within the collagen-induced arthritis (CIA) model, and the antagonist BiP peptide (456–475) is further effective in reversing the agonist BiP peptide’s (336–355) capacity to induce T cell proliferation in vitro [55]. This tolerogenic effect is common and also reported with peptides from mycHSP70/DNAk, mycHSP60/GroEl2, and pan-DR HSP60 [58,62,95]. As a consequence, atypical mycobacteria such as M. avium can be found in RA patients’ lungs [96], and RA patients present an elevated risk of mycobacterial infections, exacerbated under anti-TNF-α therapies [97]. The dual role of DRB1-SE’s restricted peptides has led to the development of competitive peptide analogs to dampen T cell activation in RA, and some of them are in clinical development, as described below [98].

#### 4.2.2. DRB1-SE Synovial Native Antigens and Degradation Products

Among notable autoantigens found in synovial articulations, collagens, which are one of the main constituents of cartilage, and in particular type 2 collagen (Col2), have demonstrated a pathogenic role in the CIA mouse model [99]. In humans and in opposition to mice, a weak T cell reactivity towards Col2 (259–273) is reported in its native form as compared to a strong activation in its galactosylated form among patients with DRB1-SE*04:01 [67]. The MHC–Col2 interaction is similar when comparing RA patients and healthy subjects carrying the DRB1*04:01 allele; however, the difference for RA patients resulted in a higher IFN-γ, IL-17, and IL-2 T cell response and an antigen specific repertoire that is correlated with clinical activity [66,69,70], and this effect is higher in early RA than in established RA [68].

Highly abundant in the extracellular matrix from synovial tissues, cartilage proteoglycans (CPG) are not directly immunogenic as the native form induces a low T cell-stimulation response. However, and particularly during inflammation, CPG fragments can be released from proteolysis, and portions of them (CPG 16–39 and 263–282) can trigger T cell proliferation, T cell activation (CD69), and cytokine release [62,100]. Other released synovial autoantigens overexpressed in RA are effective in inducing a proliferative T cell response with pro-inflammatory cytokine production; this list included the heterogeneous nuclear ribonucleoprotein A2 (hn-RNP-A2/RA33), G1 aggrecan, and the peptidylarginine deiminase (PAD) 4 [74,75,76].

Using a structural approach, Col2 261–273 (AGFKGEQGPKGEP) presented a homology with the virulence-associated trimeric transporter (Vta 755–766) from *Haemophilus parasuis* (AGPKGEQPKGE) and CPG 263–282 (YLAWQAGMDMCSAGW) with the Yersinia outer membrane protein (Yop) 68–82 from *Yersinia* sp. (QKQLGWQAGMDEART) [71,100]. According to molecular mimicry theory, these two bacterial peptides, Vta 750–766 and Yop 68–82, have further demonstrated their capacity to induce a T cell response similar to their human counterparts in RA patients.

#### 4.2.3. DRB1-SE Restricted Antigens and Post-Translational Modifications

PTM and in particular the conversion of arginine to citrulline generates “altered-self” peptides that become electropositive, allowing for insertion in the electronegative P4 pocket of the DRB1-SE, whereas the electropositive P4 pocket of the RA-protective/resistant DRB1-SE*04:02 compromises this insertion [101]. Therefore, citrullination expands the repertoire of peptides from the extracellular matrix and cellular components, and these peptides may interact with DRB1-SE and further enhance the T cell response. This effect of citrullinated peptides on T cells is suspected to result from a direct contact with the TCR from autoreactive CD4+ T cells rather than a higher affinity to the MHC, as initially stated for Col2 [79]. The citrullination process that is catalyzed by PAD4, an enzyme with increased activity during inflammation, cell death, and stress, has been reported in joints in RA patients [102] and in lung epithelial cells in tobacco smokers. Consistent with this is a report that smoking increases the risk of developing ACPA among DRB1-SE positive individuals [103].

The comparative analysis of the T cell response between non-citrullinated and citrullinated (Cit) peptides recognized by ACPA (fibrinogen, vimentin, α-enolase, Col2, aggrecan, human cartilage (HC), HCgp39, tenascin, and cartilage intermediate layer protein (CILP)) revealed [77,78,80,83,84,85,86,89] (i) that citrullinated peptides induce a T cell proliferation that can be associated with an activation when testing the native counterparts or not, as reported for aggrecan and tenascin; (ii) that T cell response is related to the presence of DRB1-SE (*04:01-04-05, *01:01-02, and/or *10:01); (iii) that a biased TCR usage was reported for tenascin 1012–1026; (iv) that differences in cytokine production are observed between peptides (elevated IL-6 levels with Cit-aggrecan and elevated IL-17 levels with Cit-fibrinogen); and (v) that Cit-peptide response number increases with established RA.

## 5. Autoreactive CD4+ T Cells

### 5.1. Phenotype (Multimers)

Class II multimers that covalently couple immunodominant peptides to DRB1-SE can be used to stain autoreactive CD4+ T cells. This strategy was applied to characterize RA-associated autoreactive CD4+ T cells by using tetramers containing TCR high affinity citrullinated peptides derived from enolase, aggrecan, fibrinogen, CILP, and vimentin. These high-affinity autoreactive CD4+ T cells are expanded in RA patients, which is not the case when using low-/medium-affinity tetramers harboring native immunodominant peptides (e.g., Col2 and HCgp39) [59,60,73]. This supports the concept that an elevated MHC-TCR affinity is required to break tolerance [82]. Moreover, high-affinity autoreactive CD4+ T cell expansion is related to memory T cells, and their level is correlated with disease activity and ACPA level [84,88,101]. High-affinity memory CD4+ T cells are oligoclonal and polarized into Th17 cells, while Th1 cells decline, and this pool is controlled by anti-TNF-α biotherapies [81].

### 5.2. DRB1-SE Peptide Repertoire

The peptide repertoire or immunopeptidome recognized by autoreactive CD4+ T cells can be appreciated by mass spectrometry following MHC elution in an unbiased manner. This strategy was applied first in a pioneering study conducted in 1995, in which 14 HLA-DR endogenous peptides were eluted from the spleen of an RA patient with Felty syndrome, and among them, a peptide derived from the human serum albumin peptide (106–120) was effective for binding DRB1-SE*04:01 [90]. Next, in 2011, two RA patients harboring DRB1-SE (*01:01 and *04:01) or not (*04:02 and *11:04) were selected, and 166 synovial peptides were eluted from DRB1 showing common peptides derived from intracellular, membrane, extracellular, and plasma proteins [104]. Some of the proteins corresponding to the 166 synovial DRB1 eluted peptides were recognized by RF (immunoglobulins) or ACPA (vimentin, fibrinogen, fibronectin, and collagen). More recently, Maggi et al. characterized synovial peptides presented by DRB1-SE from three RA patients and reported that the corresponding parental proteins were increased in the synovial fluid from RA patients (20%), recognized by autoantibodies (10%), and/or known to elicit an RA T cell response (5%) [79]. Next, authors reported the capacity of both native (gelsoline, histone H2B/H4, myeloperoxidase (MPO)) and citrullinated peptides (Histone H2B and proteoglycan 4) to trigger IFN-γ and CD40L expression on circulating CD4+ T cells from 29 RA patients as compared to 12 healthy controls. 

*Prevotella copri* is considered a pathogenic taxon in RA. Indeed, gut microbiome variations (dysbiosis) including *P. copri* are associated with the presence of DRB1-SE in healthy controls [105], with RA at early stage [106], but also with other metabolic syndromes such as insulin-resistant type II diabetes [107]. In synovial fluid from an RA patient harboring two copies of DRB1-SE, the detection of *Prevotella* sp. 16S rRNA was associated with the isolation of the HLA-DR-restricted peptide from a 27 kDa protein of *P. copri* (Pc-27) in peripheral blood mononuclear cells (PBMC) [91]. Two other self-peptides derived from N-acetylglucosamine-6-sulfatase (GNS) and filamin A (FLNA) sharing 67% and 80% homology with *P. copri*, respectively, were further eluted from the synovium [92]. In addition, the same team reported four new HLA-derived peptides from *P. copri* that were identified in new RA patients [93]. *P. copri*-derived peptides and self-GNS and -FLNA peptides were effective for inducing a Th1 response with IFN-γ release in PBMC from RA patients at new onset, and IgA *P. copri* antibody levels were correlated with ACPA values.

### 5.3. T and B Cell Epitopes

The development of ectopic or extrafollicular germinal centers occurs in RA synovial tissues and drives immune responses. In these structures, B cells contribute to T cell activation through the expression of MHC class II and co-stimulatory molecules (APC role), to pro-inflammatory cytokine production, and to the local production of AMPA and RF. In this context, the simultaneous activation of B-cells and T-cells can be promoted by overlapping and shared immunogenic epitopes. As reported in Figure 2, among the long list of T and B cell antigens associated with RA, a limited number of epitopes are recognized by the two cell types: Cit-tenascin [87], HSP40/DNAj [65], PAD4 [76], and vimentin [77]. Then, these major overlapping T/B-cell epitopes can amplify B cell-dependent antigen presentation (via MHC), autoantibody production, and autoreactive CD4+ T cell selection and expansion. Accordingly, T/B cell cooperation in RA was illustrated in the “hapten carrier model” that suggests that B cells harboring DRB1-SE present PAD4 peptides to T cells, which in turn help ACPA-producing B cells. In this model PAD4 is the carrier and citrullinated peptides are the haptens [76].

### 5.4. Therapeutic Peptides

Altered peptide ligands (APL) were initially developed as controls for CD4+ T cell-immunodominant peptides, but it was quickly observed that APL were effective in (i) repressing proliferation without altering cytokine and antibody production by fostering T-B cell cooperation, (ii) promoting Treg development instead of T helper polarization, or (iii) acting as a super-agonist, as recently reviewed by Candia et al. [108]. Then the possibility of creating APL opens up new and attractive therapeutic options for RA, and for that purpose, a three-step strategy has been initiated. First, APL has been screened in vitro to test partial agonist effects, as observed following amino acid substitution in Col2 (256–271) and in HCgp39. Both peptides conserved DRB1-SE-binding capacity but failed to induce a T cell response [109,110]. Another mechanism is described for HSP60 E18 mutants (90–109, HSP60 APL1:L109, Hsp60 APL2:L103) that are effective in both increasing the proportion of Treg and suppressing IL-17 level, and APL-1 is further effective in inducing apoptosis in autoreactive CD4+ CD25+ T cell and increasing IL-10 levels when incubated with PBMC from RA patients [111,112,113] (Figure 3). Second, APL’s capacity to reverse the disease has been tested in animal models. To this end, the synthetic Col2 (245–270, A260, B261, N263) is effective in suppressing CIA directly by increasing the Th2 response toward Th1/Th17 responses or indirectly when APC-trained T cells are passively transferred [114,115,116]. Both HSP60 APL-1 and APL-2 mutants reduce adjuvant arthritis in ill rats treated with *Mycobacterium tuberculosis* [112,113]. Similarly, the prophylactic use of APL-12 derived from human glucose 6 phosphate isomerase (hGPI 325–339) improves the severity of arthritis via Treg induction in the GPI-induced arthritis mouse model [117]. Finally, and third, phase I clinical trials have started, conducted for HSP60 APL-1 (CIGB-814) in order to evaluate its pharmacological characteristics, and good tolerance with clinical improvement has been reported in 20 RA patients [118,119].

## 6. Conclusions

RA is a chronic immune and inflammatory disorder for which no cure exists, and this results from a lack of understanding of the mechanisms leading to T cell hyperactivation, which in turn initiates and sustains the disease. Analysis of the peptides leading to T cell hyperactivation supports a complex process that includes a combination of disease-provoking microorganisms, genetic mutations, inflammation, and environmental factors. The common denominators are molecular mimicry, neoantigens, MHC restriction, TCR engagement, and overlapping T/B epitopes. Consequently, a better characterization of the hyperactivated autoreactive T cells in RA allows for personalized perspectives regarding disease prediction, disease follow-up, therapeutic response, and the development of new therapeutics to prevent and control the development hyperactivated autoreactive T cells.

## Figures and Tables

**Figure 1 ijms-24-04296-f001:**
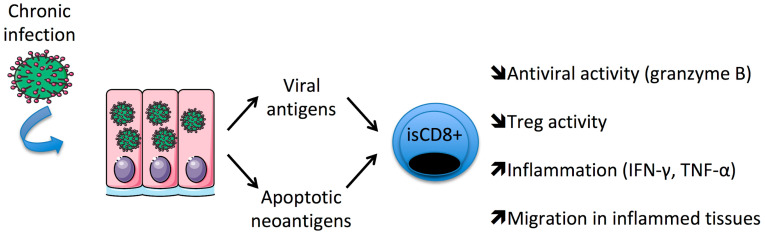
Persistent/chronic infections favor the expansion of immunosenescent (is)CD8+ T cells in patients with rheumatoid arthritis.

**Figure 2 ijms-24-04296-f002:**
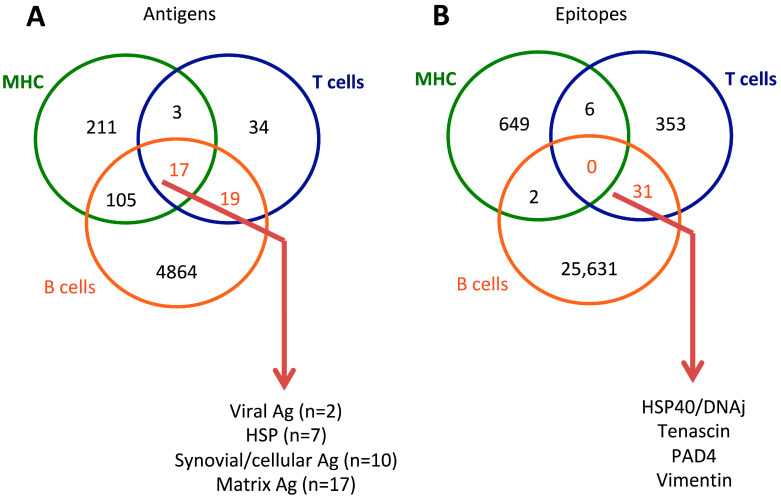
Venn diagram of antigens (**A**) and epitopes (**B**) reported in patients with rheumatoid arthritis (RA) from the Immune Epitope Database (https://www.IEDB.org, accessed on 7 February 2023) that reports T cells, B cells, and major histocompatibility (MHC) assays.

**Figure 3 ijms-24-04296-f003:**
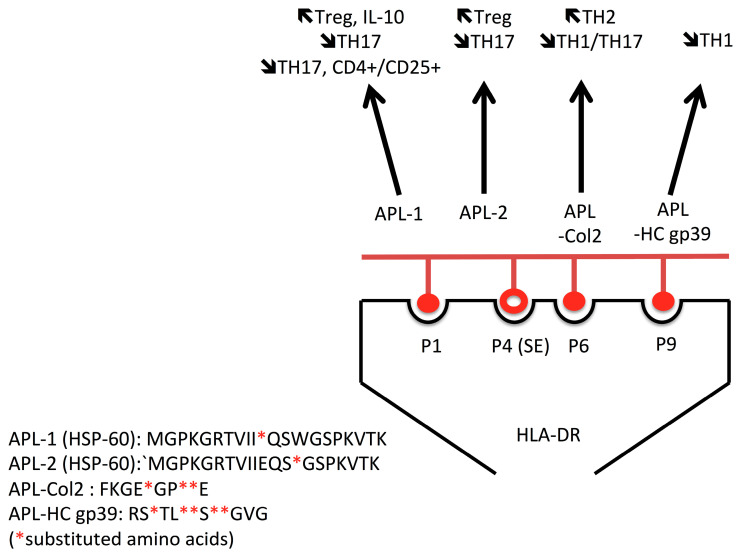
Several antigens, including heat shock proteins (HSP) 60, collagen 2 (Col2), and human cartilage glycoprotein (gp)39, are immunogenic and associated with disease onset. Altered peptide ligands from these antigens are effective to shift the pro-inflammatory (TH1/TH17) cellular immune response into an anti-inflammatory (TH2) and or regulatory (Treg) T cell response.

**Table 1 ijms-24-04296-t001:** RA-associated information extracted from the Immune Epitope Database and analysis resource website (http://www.iedb.org, accessed on 7 February 2023). Abbreviations: MHC: major histocompatibility complex; TCR: T cell receptor.

	T Cells	MHC Elution	B Cell
Assays	865	859	26,539
Epitopes	1390	657	25,664
Antigens	79	366	5100
TCR	153	0	20
References	103	5	235

**Table 2 ijms-24-04296-t002:** Techniques used to evaluate and characterize T cells responses against microbial and host epitopes. Abbreviations: ELISpot: enzyme-linked immunospot; ELISA: enzyme linked immunosorbent assay; MHC: major histocompatibility complex; TCR, T cell receptor; RT-PCR: reverse transcription polymerase chain reaction, NGS: next generation sequencing.

T Cell Exploration	Type	Assays
T cell activation	Proliferation	3H thymidine Optical density
	Cytokines and chemokines	ELISpot, ELISA Flow cytometry Multiplex (Luminex)
T cell phenotype	Characterization, cell sorting	MHC multimers
TCR	TCR repertoire (oligoclonality)	Multiplex RT-PCR, NGS

**Table 3 ijms-24-04296-t003:** Main human leukocyte antigen (HLA) class I antigens reported in the RA- associated immune epitope database (IEDB). Abbreviations: HLA: human leukocyte antigen; SF: synovial fluid; IEDB: immune epitope database; Ref: reference; CMV: cytomegalovirus; EBV: Epstein–Barr virus; BMLF1: Epstein–Barr virus lytic cycle protein; BZLF1: Epstein–Barr virus immediate-early protein; EBNA3A: Epstein–Barr virus latent nuclear antigen 3A; IAMP: influenza A matrix protein; MYH9: myosin-9; VIME: vimentin; ACTB1: actin cytoplasmic 1.

Organism	Antigen	Main Epitope	HLA Restriction	T Cell Effect	T Cell Subset (Repertoire)	Blood/SF [Ref]
CMV	pp65	NLVPMVATV	A2	Enrichment (SF > blood)	CD8+	SF, blood [30,31,32,33]
EBV	BMLF1	GLCTLVAML	A2	Enrichment (SF > blood)	CD8+ (Vβ2/12/16)	SF, blood [32,33,34]
	BZLF1	RAKFKQLL	B8, B61, Cw1	Enrichment (SF > blood)	CD8+	SF, blood [32,33,34]
	EBNA3A	FLRGRAYGL	B8, B61	No enrichment (SF = blood)	CD8+	SF, blood [32,33,34]
Influenzae	IAMP	GILGFVFTL	A2	No enrichment (SF = blood)	CD8+ (Vβ17)	SF, blood [33]
Apoptotic (human)	MYH9	QLFNHTMFI VLMIKALEL	A2	Autoreactive Teff (RA > HC)	CD8+	Blood [35,36,37]
Apoptotic (human)	VIME	LLQDSVDFSL SLQEEIAFL	A2	Autoreactive Teff (RA > HC)	CD8+	Blood [35,36]
Apoptotic (human)	ACTB1	FLGMESCGI	A2	Autoreactive Teff (RA > HC)	CD8+	Blood [35,36]

**Table 4 ijms-24-04296-t004:** Human leukocyte antigen (HLA) class II antigens reported in the RA-IEDB data and the literature. Abbreviations: Activ: activation assay; BiP: Binding immunoglobulin protein; CILP: cartilage intermediate-layer protein; Col2: collagen 2; FLNA: filamin A; GNS: N-acetylglucosamine-6-sulfatase; Hae: haemagglutinin; HC: healthy controls; Hcgp39: human cartilage glycoprotein 39; hn-RNP-A2: heterogeneous nuclear ribonucleoprotein A2; HSA: human serum albumin; HSP: heat shock protein; MMP-3: metalloproteinase 3; MP1: influenzae matrix protein1; MPO: myeloperoxydase; NT: not tested; *P. copri:* Prevotella copri; PAD4: peptidylarginine deiminase 4; PG4: proteoglycan 4; Prolif: proliferation assay; RA: rheumatoid arthritis; SF: synovial fluid; VIME: vimentin; Vta: virulence-associated trimeric autotransporter; Yop: Yersinia outer membrane protein; *: HLA gene variants when known.

Organism	Antigen	Main Epitope (Antagonist, B Cell Epitope)	HLA Restriction	Native Form Effect on T Cells	If Not Specified, Citrullinated Form Effect	Blood/SF [Reference]
Peptides derived from heat shock proteins and conserved proteins						
Mycobacteria	HSP70/DNAk (287–306)	DRTRKPFQSVIADTGISVSE	DR4	Prolif (RA > HC), IFN-γ, IL-17	-	Blood [55]
Mycobacteria	LpqH (31–50)	SGETTTAAGTTASPGAASGPK	DRB1-SE	Prolif (RA > HC), IFN-γ		Blood [56,57]
Human	BiP (336–355)	RSTMKPVQKVLEDSDLKKSD	DR4	Prolif (RA > HC), IFN-γ, IL-17	-	Blood [58]
Influenzae	HAE (307–319)	PKYVKQNTLKLAT	DR4	Prolif (RA > HC), IFN-γ	-	Blood [56,59,60]
Influenzae	MP1 (17–29)	SGPLKAEIAQRLE	DR4	Prolif (RA > HC), IFN-γ	-	Blood [56,59,61]
Human	BiP (456–475)	DNQPTVTIKVYEGERPLTKD (antagonist)	DR4	Weak prolif, Treg (IL-10)	-	Blood [55]
Mycobacteria	HSP65/GroEL2 (256–270)	ALSTLVVNKIRGTFK	DRB1-SE	Prolif (RA > HC), IL-1β, IL-6, TNF-α	-	Blood [62]
Human	HSP60 (535–549)	ALSTLVLNRLKVGLQ	DRB1-SE	Prolif (RA > HC), IL-1β, IL-6, TNF-α	-	Blood [62]
Mycobacteria/human	HSP60	panel (antagonist)	pan-DR	Prolif (RA = HC), IL-10 > TNF-α	-	Blood [62]
Rat	MMP3 (446–460)	FFYFFTGSSQLEFDP	DRB1	Prolif (RA > HC), IL-4, IL-1β, TNF-α	-	Blood [63]
Mycobacteria	HSP40/DNAj	QKRAAYDQYGHAAFE (B cell)	DRB1-SE	Prolif (RA > HC)		Blood [64,65]
Synovial peptides derived from native and cross-reactive proteins						
Human	Col2 (259–273)	GIAGFKGEQGPKGET	DR4	Weak prolif (RA > HC), IFN-γ, IL-17 and IL-2	Galactosylation: strong prolif, IFN-γ, IL-17 and IL-2	Blood/SF [66,67,68,69,70]
Human	Col 2 (261–273)	AGFKGEQGPKGET	DR4	Activation (RA > HC), CD40L, IL-17, IL-4	Activation (RA > HC), CD40L, IL-17, IL-4	Blood [66]
*Haemophilus parasuis*	Vta (755–766)	AGPKGEQPKGE	DR4	Prolif, IL-17	-	Blood [71]
Human	Cartilage proteoglycan (268–282)	YLAWQAGMDMCSAGW	DRB1-SE	Prolif (RA > HC), IL-1, IL-2, IL-6, IL-10, TNF-α, MCP1	-	Blood [62]
Human	HCgp39 (263–275)	FTLASSETG	DR4	Prolif (RA > HC)	-	Blood, SF [72,73]
*Yersinia* sp.	Yop (68–82)	QKQLGWQAGMDEART	DRB1-SE	Prolif	-	Blood [62]
Human	hnRNP-A2/RA33 (117–133)	RDYFEEYGKIDTIEIIT	DRB1-SE	Prolif (RA > HC), IFN-γ, IL-2		Blood [74]
Human	G1 aggrecan	pool	HLA-DR	Prolif (RA > HC), IFN-γ, TNF-α	-	Blood/SF [75]
Human	PAD4	DPGVEVTLTMKAASGSTGDQ (B cell)	DRB1-SE	Prolif (RA > HC), TNF-α	-	Blood [76]
Extracellular-matrix and other peptides with citrulline modifications (R underline)						
Human	Fibrinogen-α (79–91)	QDFTNRINKLKNS	DRB1-SE	No response	Weak prolif, IL-17 > IFN-γ, IL-6 (RA > HC)	Blood [77]
Human	Col2 (1237–1249)	QYMRADQAAGGLR	DRB1-SE	No response	Weak prolif, IFN-γ, IL-6, IL-17 (RA > HC)	Blood [77,78,79]
Human	Col2 (311–325)	APGNRGFPGQDGLAG	DRB1-SE *10:01	No response	Activ. (CD40L), TNF-α, IL-17F, IL-10, IL-13	Blood [78]
Human	VIME (66–78)	SAVRARSSVPGVR (B cell)	DRB1-SE	No response	Weak prolif, IFN-γ, IL-6, IL-17 (RA > HC)	Blood [77,80,81,82]
Human	Aggrecan (84–103)	VVLLVATEGRVRVNSAYQDK	DRB1-SE	No response	Weak prolif, IL-6 > IFN-γ, IL-17 (RA > HC)	Blood [77,80,83,84]
Human	Fibrinogen	Pool	DRB1	Prolif similar to controls	Prolif similar to controls	Blood [85]
Human	Tenascin	VSLISRRGDMSSNPA (B cell)	DRB1-SE *04:01	No response	Weak prolif (RA > HC), IFN-γ > IL-17, IL-10	Blood, SF [86,87]
Human	Tenascin 56	QGQYELRVDLRDHGE (B cell)	DRB1-SE *04:01	No response	Weak prolif (RA > HC), IFN-γ > IL-17, IL-10	Blood, SF [86,87]
Human	CILP (982–996)	GKLYGIRDVRSTRDR	DRB1-SE *10:01	No response	Prolif, IL-17	Blood [59,81,88]
Human	α-enolase (26–40)	TSKGLFRAAVPSGAS	DRB1-SE	No response	T cell response (RA > HC), IFN-γ	Blood [59,89]
Human	α-enolase (326–340)	KRIAKAVNEKSCNCL	DRB1-SE	-	Enrichment (SF > blood), Prolif, IL-17	Blood, SF [59,60,81]
HLA-DR peptides eluted from synovial tissues with citrulline modification (R underline)						
Human	HAS (106–120)	RETYGEMADCCAKQEPE	DRB1-SE *04:01	-	-	Spleen [90]
*P. copri*	Pc-27 (2–19)	KRIILILTVLLAMLGQVAY	DRB1-SE	IFN-γ release (RA > HC)	-	Blood [91,92]
Human	GNS	FEPFFMMIATPAPH	DRB1-SE	IFN-γ release (RA > HC)	-	SF [91,92,93]
Human	FLNA	NPAEFVVNTSNAGAG	DRB1-SE	IFN-γ release (RA > HC)	-	SF [91,92,93]
Human	Gelsolin	DAYVILKTVQLRNGN	DRB1-SE *01:01	Activ (CD40L), IFN-γ	No effect	SF [79]
Human	Histone H2B	MNSFVNDIFERI	DRB1-SE *04:01	Activ (CD40L), IFN-γ	Activ (CD40L), IFN-γ, TNF-α	SF [79]
Human	Histone 4	DNIQGITKPAIRR	DRB1	Activ (CD40L), IFN-γ	-	SF [79]
Human	PG4	THTIRIQYSPAR	DRB1-SE *04:01	No effect	Activ (CD40L), IFN-γ, TNF-α	SF [79]
Human	MPO	SNEIVRFPTDQLTPDQ	DRB1	Activ (CD40L), IFN-γ	-	SF [79]

## Data Availability

Data is contained within the article.

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
