# Peer review of "Characteristics of the (Auto)Reactive T Cells in Rheumatoid Arthritis According to the Immune Epitope Database"

_ijms, 2023, doi:10.3390/ijms24054296_

Round 1

Reviewer 1 Report

The authors did a review on the T-cell epitopes and their outcomes as a cellular response in RA using IEDB database. Although any contribution in the RA field is important. This review does not add much to the existing knowledge in the field. 

1. To begin with it is not clear how the data was processed once extracted from the IEDB database for the downstream information evaluation.

2. Line 89 - 1390? Can't see this number in Table 1.

3. From 2.1-3.4, we read just the basics of the techniques and approaches and somewhere the manuscript loses its focus on relating the epitopes identified through IEDP to those approaches except slightly in viral reactivation.

4. The method in this review is not very clear. Did the authors download all RA associated peptide epitopes and then performed their own evaluation OR are the authors describing the collated results about the categorized antigens/epitopes through their own literature review?

5. Similar to point #2, the abstract and the manuscript content does not match. From the abstract, it seems that one will be reading from about the T-cell epitopes in the manuscript and that description does not match with the second half the manuscript either with section #4. I would recommend authors to reconsider editing their manuscript based on the focus of their goals.

Author Response

Reviewer 1:

The authors did a review on the T-cell epitopes and their outcomes as a cellular response in RA using IEDB database. Although any contribution in the RA field is important. This review does not add much to the existing knowledge in the field. 

#1 : To begin with it is not clear how the data was processed once extracted from the IEDB database for the downstream information evaluation. The method in this review is not very clear. Did the authors download all RA associated peptide epitopes and then performed their own evaluation OR are the authors describing the collated results about the categorized antigens/epitopes through their own literature review?

Answer R1A : we apologize for being imprecise regarding the methodology used in this review to achieve our main objective, which is the characterization of the (auto)reactive T cell subset in rheumatoid arthritis through the use of the IEDB database (title changed accordingly). For that, we have proceeded in 3 steps :

  • Step 1, the IEDB database was generated and the table was added as a supplementary data file. This list recapitulates all peptides tested in the context of RA, however this list can’t be used per se as only a subset is able to discriminate RA-associated T cell specific epitopes from non-specific epitopes, which needs to be done manually in depth by reading all manuscripts.
  • Step 2, from the IEDB manuscript analysis we next manually established a list of major epitopes/antigens that were subdivided into two groups according to their presentation by HLA class I (table 3) and class II (Table 4).
  • Step 3, from IEDB publications and database, tables 3&4 were updated in order to present RA-associated T cell characteristics according to the related peptides. For that several items of information were collected : main epitope and related antigen, HLA restriction, effect on T cells, comparison RA/controls, T cell subsets, cellular origin, and references. For this revision, tables 3 and 4 were checked and updated when necessary.

Accordingly, this section was added, thanks for the comment:

« From this list and since not all epitopes and antigens are relevant to characterize (auto)reactive T cells in RA, publications were individually reviewed in depth in order to establish a list of major epitopes/antigens based on their relevance to RA. Moreover and when available, information regarding HLA class I/II restriction, T cell activation, the phenotype of the T cell population expanded, peripheral blood or synovial fluid T cell origin, capacity to enter the docking site of the DRB1-SE, cross-reactivity, importance of PTM in generating neo-epitopes, antagonist effect, and TCR molecule usage were further collected from the database and/or from publications. From these sources, tables 3 and 4 were generated based on HLA I/II restriction usage and antigen characteristics for the latter. To help the reader, main techniques used to characterize RA-associated (auto)reactive T cells are reviewed below. »

#2 : Line 89 - 1390? Can't see this number in Table 1.

Answer R1B : the typo was corrected, thanks for the comment.

#3 : From 2.1-3.4, we read just the basics of the techniques and approaches and somewhere the manuscript loses its focus on relating the epitopes identified through IEDP to those approaches except slightly in viral reactivation. 5. Similar to point #2, the abstract and the manuscript content does not match. From the abstract, it seems that one will be reading from about the T-cell epitopes in the manuscript and that description does not match with the second half the manuscript either with section #4. I would recommend authors to reconsider editing their manuscript based on the focus of their goals.

Answer R1C : In order to clarify the purpose and to focus on our main goal : Characteristics of the (auto)reactive T cells in rheumatoid arthritis according to the Immune Epitope Database (new title), substantial changes have been made in the abstract. Thanks for your comment.

Reviewer 2 Report

This manuscript deals with the analysis of T cell epitopes and cellular responses in rheumatoid arthritis (RA), This analysis is performed looking at the immune epitope database.

The manuscript is well organized and written.

Also, it is of interest the last paragraph regarding the altered peptide ligands and their potential use for RA treatment. It would be of help for the reader to explain in an additional figure/cartoon the possible application of these APL alone or together to conventional RA therapy.

Author Response

This manuscript deals with the analysis of T cell epitopes and cellular responses in rheumatoid arthritis (RA), This analysis is performed looking at the immune epitope database.

The manuscript is well organized and written.

Also, it is of interest the last paragraph regarding the altered peptide ligands and their potential use for RA treatment. It would be of help for the reader to explain in an additional figure/cartoon the possible application of these APL alone or together to conventional RA therapy.

Answer R2: Thanks for your comment and according to your suggestion a figure was added.

Reviewer 3 Report

Rheumatoid arthritis (RA) is a painful disease that does not have any cure yet. The limited knowledge about the cause and progression mechanism of this disease restricts the drug discovery research to find a permanent treatment for RA. The current review article discussed the role of the T cell hyperactivation and the different cascade effects that have been discovered in the last one decade or so help the propagation of the disease. The current article is important in this context. The article is well written within the scope of this publication format. The available research and data till date were compiled in a comprehensive manner to explain how T cell activation and differentiation leads to propagation of this inflammatory autoimmune disease. This article will be a valuable addition to the literature and help design future research to find a permanent treatment for RA.

Hence I recommend the article to be published in its current form.

Author Response

Rheumatoid arthritis (RA) is a painful disease that does not have any cure yet. The limited knowledge about the cause and progression mechanism of this disease restricts the drug discovery research to find a permanent treatment for RA. The current review article discussed the role of the T cell hyperactivation and the different cascade effects that have been discovered in the last one decade or so help the propagation of the disease. The current article is important in this context. The article is well written within the scope of this publication format. The available research and data till date were compiled in a comprehensive manner to explain how T cell activation and differentiation leads to propagation of this inflammatory autoimmune disease. This article will be a valuable addition to the literature and help design future research to find a permanent treatment for RA.

Hence I recommend the article to be published in its current form.

Answer R3: Thanks for your positive comments regarding our manuscript, we really appreciate.

Round 2

Reviewer 1 Report

Thank you for addressing the comments in the manuscript.